# Validation of the Polar V800 heart rate monitor and comparison of artifact correction methods among adults with hypertension

Burak Cilhoroz[1,2]*, David Giles[3⊙], Amanda Zaleski[1,4‡], Beth Taylor[1,4‡], Bo Fernhall[5⊙], Linda Pescatello[1,6⊙]

1 Department of Kinesiology, University of Connecticut, Storrs, Connecticut, United States of America, 2 Department of Exercise Science, Syracuse University, Syracuse, New York, United States of America, 3 Health and Social Care Research Centre, University of Derby, Derby, United Kingdom, 4 Department of Preventive Cardiology, Hartford Hospital, Hartford, Connecticut, United States of America, 5 College of Applied Health Sciences, University of Illinois at Chicago, Chicago, Illinois, United States of America, 6 Institute for Systems Genomics, University of Connecticut, Storrs, Connecticut, United States of America

⊙ These authors contributed equally to this work.
‡ These authors also contributed equally to this work.
* burak.cilhoroz@uconn.edu

**Data Availability Statement:** All relevant data are within the paper and its Supporting Information files.

## Abstract

Heart rate variability (HRV) measurements via ambulatory monitors have become common. We examined the validity of recording R-R intervals using the Polar V800™ compared to 12-lead electrocardiograms (ECG) among middle-aged (44.7±10.1years); overweight to obese (29.8±4.3 kg.m$^{-2}$) adults (n = 25) with hypertension (132.3±12.2/ 84.3±10.2 mmHg). After resting for 5-min in the supine position, R-R intervals were simultaneously recorded using the Polar V800™ and the 12-lead ECG. Artifacts present in uncorrected (UN) R-R intervals were corrected with the Kubios HRV Premium (ver. 3.2.) automatic (AC) and threshold-based (TBC) correction, and manual correction (MC) methods. Intra-class correlation coefficients (ICC), Bland-Altman limits of agreement (LoA), and effect sizes (ES) were calculated. We detected 71 errors with the Polar V800™ for an error rate of 0.85%. The bias (LoAs), ES, and ICC between UN and ECG R-R intervals were 0.69ms (-215.80 to +214.42ms), 0.004, and 0.79, respectively. Correction of artifacts improved the agreeability between the Polar V800™ and ECG HRV measures. The biases (LoAs) between the AC, TBC, and MC and ECG R-R intervals were 3.79ms (-130.32 to +137.90ms), 1.16ms (-92.67 to +94.98ms), and 0.37ms (-41.20 to +41.94ms), respectively. The ESs of AC, TBC, and MC were 0.024, 0.008, and 0.002, and ICCs were 0.91, 0.95, and 1.00, respectively. R-R intervals measured using the Polar V800™ compared to 12-lead ECG were comparable in adults with hypertension, especially after the artifacts corrected by MC. However, TBC correction also yielded acceptable results.

**Funding:** This study was supported by funding from the University of Connecticut Institute for Collaboration on Health, Intervention, and Policy (InCHIP) and the United States Department of Agriculture (SAES,HATCH Project No. CONS00954). Both funding were awarded to LSP. The funders had no role in study design, data collection and analysis, decision to publish, or preparation of the manuscript.

**Competing interests:** The authors have declared that no competing interests exist.

## Introduction

Hypertension is the most common, costly, and preventable cardiovascular disease risk factor [1] in the US with nearly one in two American adults having hypertension [2]. Modifiable (e.g., obesity, physical inactivity, smoking, dyslipidemia) and non-modifiable (e.g., age, gender, family history) risk factors can increase the likelihood of developing hypertension [2]. However, the underlying etiology for the development of hypertension is largely unknown. One of the hypotheses regarding the initiation, progression, and maintenance of hypertension is alterations in the neural control of blood pressure (BP) [3]. These neural alternations generally emerge as excessive sympathetic stimulation and parasympathetic inhibition of the autonomic nervous system (ANS) [4]. The sinoatrial node integrates the inputs from the ANS to adjust heart rate (HR) in response to the constantly changing internal and external environment to maintain homeostasis [4]. This adjustment causes an oscillatory pattern in the HR resulting from beat-to-beat fluctuations in the time period between sequential heartbeats, termed heart rate variability (HRV). HRV is used to non-invasively measure ANS modulation.

Due to the recent technological advances in signal analysis, users can now utilize accessible, simple, and affordable heart rate (HR) monitors to measure HRV rather than the gold standard criterion of the 12-lead electrocardiogram (ECG). Indeed, a PubMed search showed the number of publications associated with HRV rose from 393 in 1995, the year of the first Polar HR monitor (Polar Vantage NV™) for measurement of HRV, to 1726 in 2019. Nevertheless, artifacts (i.e., random distortion of usual R-R intervals) with technical (e.g., excessive noise which causes incorrect detection of R peak) and physiological (e.g., ectopic beat) origins can substantially impair the ability of HR monitors to accurately detect R-R intervals that are used to calculate the time, frequency, and non-linear domains of HRV [5–7].

Adults with hypertension may have a higher number of artifacts in HRV recordings than healthy individuals. Low R-wave amplitude may be observed in patients with the early stages of hypertension before left ventricular hypertrophy develops, a typical response to chronic high BP [8]. Further, increased chest wall size and epicardial adipose tissue can lead to diminished R-wave amplitude among patients with hypertension since more than 80% of this population are overweight to obese [9]. Moreover, arrhythmias (e.g., premature ventricular contractions) can disrupt R-wave morphology leading to HRV recording errors. Arrhythmias are common among hypertensives due to the disturbances in sinus and atrioventricular nodes and electrolyte imbalances caused by antihypertension medications (e.g., diuretics) [10]. Since HRV is measured using R-R intervals, low R-wave amplitude and distorted R-wave morphology can affect HR monitors' ability to correctly identify R-R intervals. Thus, correcting the artifacts may be even more important in this clinical population to acquire valid HRV measures and accurately interpret the autonomic status of these patients.

The manual correction (MC) of artifacts with visual identification of the R-R intervals combined with the choice of a proper correction method is the most accurate method for obtaining valid HRV measures [6, 7]. However, the MC is laborious and requires multiple individuals and long hours of training to properly utilize. Various HRV software packages, therefore, offer easy-to-use automatic correction methods to edit artifacts, making them more attractive than the MC. Kubios is the most popular HRV software that is used to automatically correct artifacts [11]. The most recent version of the software (Kubios HRV Premium [ver. 3.2]) provides two options for correcting technical and physiological artifacts, the: 1) automatic correction (AC) and 2) threshold-based correction (TBC) methods. The AC uses a time series with differences between sequential R-R intervals to identify artifacts. The TBC makes comparisons of each R-R interval against a local R-R interval average and corrects those that exceed or fall behind the local R-R interval average more than a selected threshold value [11].

Compromised autonomic modulation is a key element in the pathology of hypertension [12]. Over time, excess sympathetic activation combined with reduced parasympathetic influence may result in left ventricular remodeling and atherosclerosis due to increased inflammation potentially as a consequence of excess sympathetic activation in patients with hypertension, increasing the risk of microvascular (e.g., vascular dementia) and macrovascular (e.g., myocardial infarction) injuries [12]. In fact, significantly lower HRV (indicative of increased sympathetic activity, or decreased parasympathetic influence, or a combination of both) has been shown in studies of patients with controlled hypertension and left ventricular hypertrophy as well as in patients with uncontrolled hypertension compared to those with controlled hypertension without left ventricular hypertrophy [13–15]. Therefore, examining HRV in hypertensives with the aim of understanding autonomic function as an additional guide to treatment may be of notable clinical significance, potentially leading to improved prognosis of this population.

Typically, ECGs are used to measure HRV for analyzing the autonomic activity of clinical populations. However, patients report discomfort with ECGs due to their cumbersome and irritating adhesive electrodes and large number of leads, which may result in low patient compliance with treatment [16]. Validation of a portable, leadless, lightweight, and easy-to-use HRV monitor without skin preparation and electrode placement among adults with hypertension can help clinicians to optimize the efficacy of therapies for the treatment of hypertension and to improve patient experience. This can also improve the effectiveness of researchers collecting HRV data and evaluating the mechanisms underlying a patient's hypertension. The Polar HR monitors including S810™ [17, 18], S810i™ [19], RS800CX™ [20], RS800GX™ [21], and the current version V800™ HR monitor [22–24] have been validated among healthy individuals. HR monitors, including the latest version of Polar V800™, have not been validated in adults with hypertension. Thus, the aims of this study were to examine 1) the validity of the Polar V800™ compared to the gold standard ECG and 2) the accuracy of AC and TBC methods of Kubios HRV Premium (ver. 3.2) and MC in correcting artifacts among adults with hypertension.

## Materials and methods

### Study population

This study was approved by the University of Connecticut and Hartford Hospital Institution of Review Boards. Subjects completed written informed consent approved by the Institutional Review Boards of the University of Connecticut and Hartford Hospital. Consistent with the Institutional Review Board approved inclusion and exclusion criteria of the other studies being conducted in the laboratory from October 2016 to May 2018, subjects in this study were recruited from the surrounding community with direct mailings and posting of flyers, media advertisements, social media, previous studies, and from places of work and college campuses with the posting of flyers, listservs, class announcements, and newsletters. Adults (n = 25) with hypertension, defined as having a systolic BP SBP≥130-<160 mmHg and/or diastolic BP DBP≥80-<100 mmHg and not on antihypertensive medication, or on antihypertensive medication with an SBP<160 mmHg and DBP<100 mmHg [2] were enrolled. The subjects' physical activity levels ranged from being sedentary to regularly physically active (i.e., 150 min a week moderate or 75 min a week vigorous intensity exercise). They were free of cardiovascular, pulmonary, renal, metabolic, or other chronic diseases and depression; had not smoked for at least 6 months prior to entry; and consumed less than two alcoholic drinks daily. Subjects with a medical history of cancer-related lymphedema, those trying to gain or lose weight and

women who were pregnant, or planning to become pregnant, or were lactating were also excluded.

## Study overview

Subjects were recruited on a rolling basis from other studies conducted in the laboratory from October 2016 to May 2018 from which their baseline data were analyzed for the purpose of this study. All subjects attended one laboratory session between 6:00 and 11:00 am. Prior to the resting HRV measurement, subjects' chests were cleaned for the ECG attachment of the 12-lead ECG electrodes. Later, the chest strap (H7) of the Polar V800™ HR monitor was placed just below the pectoralis major muscles. Then, resting HRV was measured in the supine position. The R-R interval time series from ECG was stored in CASE™ (ver. 6.6) GE Healthcare system while those from Polar V800™ HR monitor was automatically stored in the Polar Flow web service.

## Procedures

Subjects were informed to consume a light breakfast 2–3 hours and abstain from drinks containing alcohol and caffeine at least 12 hr before testing. Body mass index (BMI kg.m$^{-2}$) was measured from body weight and height with a calibrated balance beam scale, and waist circumference (WC) was measured using a Gullick tape at the narrowest part of the torso [25]. Resting BP was measured following the American Heart Association's standard procedures with an automated BPTRU monitor (BPTRU Medical Devices; Coquitlam, Canada). The measurements of BMI, WC, and BP preceded the HRV recording.

The ECG electrodes were placed in the Mason-Likar configuration using the GE Stress System (CASE, Milwaukee, WI) [26]. The ECG signal was checked to ensure that it was consistent and free of noise. After the signal was confirmed to be acceptable, the Polar V800™ H7 chest strap was fitted below the pectoralis major muscles and applied as described by the manufacturer. Subjects were in the supine position for 10 min with the last 5 min recorded for the analysis in a quiet, low light, and temperature-controlled room. During the recording, subjects were instructed to relax as much as possible but not sleep, move, or talk. Breathing frequency was paced at 12 breaths min$^{-1}$ using a metronome to control for the respiratory influences on HRV measures (particularly HF or respiratory band).

Following the HRV measurement, a maximal cardiopulmonary graded exercise stress test (GEST) was performed using the Balke protocol on a Trackmaster treadmill (Full Vision, Inc., Newton, KS) to determine peak oxygen consumption (VO$_{2peak}$) which was examined in the statistical analysis for its potential influence on HRV measures.

## R-R interval processing

Initially, the 12-lead ECG and Polar V800™ HR monitor were started to simultaneously record R-R intervals at an acquisition rate of 1000 Hz. Cardiology XML files obtained from the ECG were imported into Kubios HRV Premium (ver. 3.2) (The Biomedical Signals Analysis and Medical Imaging Group, University of Kuopio, Finland) in order to export the R-R intervals detected automatically by its built-in QRS detection algorithm. Kubios HRV Premium (ver. 3.2) marked each R-wave with "+" sign that could be moved or removed to correct falsely detected R-waves. The detected R-R intervals were manually inspected on the ECG tachogram to ensure there were no false R-wave detections. If an R-wave was falsely determined, one of the authors (BTC) replaced it by moving "+" signs on the correct R-wave. The detected R-R intervals from ECG were then checked by another author (DAG) to ensure data accuracy and saved in a space delimited ASCII text file. In addition, the R-R intervals recorded by Polar

V800™ HR monitor were exported from the Polar Flow web service (ver. 2.3; Polar Electro Oy, Kempele, Finland) in a space delimited ASCII text file.

## Artifact identification

First, the R-R intervals from the ECG and Polar V800™ HR monitors were imported into the same spreadsheet side-by-side. Next, the R-R intervals from both devices were synchronized by inserting 0 ms interval to ease the comparison between the ECG and uncorrected (UN) V800™ HR monitor R-R intervals (these were subsequently removed prior to analysis). The technical artifacts from the Polar V800™ HR monitor recordings or physiological artifacts were then identified by comparing the R-R intervals from both devices. An artifact from the V800™ HR monitor was identified when the differences between ECG and Polar V800™ HR monitor R-R intervals were greater than 20 ms. Subsequently, the differences between the R-R intervals of ECG and Polar V800™ HR monitor (i.e., type of artifacts) were assigned one of seven error types of the error identification and correction guideline developed by Gamelin et al. [18] that was recently updated by Giles et al. [22] Table 1. Artifact identification was performed by the first author (BTC) and each step of this process was checked by a co-author (DAG).

## Artifact correction

Artifacts were manually and automatically corrected after their identification. All artifact corrections were performed by the first author (BTC) and each step of this process was checked by a co-author (DAG). The MC was used only for T2, T3, T4, T5, and T6b artifacts present in R-R intervals derived from the Polar V800™ following the guidelines provided in Table 1. The T1 and T6a artifacts in the Polar V800™ R-R intervals, however, were not corrected since it is not possible to identify both artifacts without simultaneous ECG recordings The corrected Polar V800™ R-R intervals by the MC were then imported into Kubios HRV Premium (ver.3.2) to calculate the HRV measures. All non-sinus beats present in the ECG signal (N = 7) were replaced by interpolated R-R intervals from adjacent R-R intervals.

**Table 1. Types, description, and correction methods of artifacts.**

| Type | Description | Correction |
|------|-------------|------------|
| T1 | A positive or negative single interval difference greater than 20 ms[a] | Artifact noted but not corrected |
| T2 | A long interval followed by a short interval while the two R-R intervals either side were less than 20 ms | Long and short intervals averaged |
| T3 | A short interval followed by a long interval while the two R-R intervals on either side were less than 20 ms | Short and long intervals averaged |
| T4 | When the HR monitor missed R-R intervals equivalent to two or three ECG R-R intervals | Artificially long R-R interval divided by the number of missed R-waves (beats)[b] |
| T5 | When the HR monitor detected extra short R-R intervals equivalent to one ECG R-R interval | Two extra short R-R intervals added together to approach corresponding ECG R-R interval |
| T6a | When the HR monitor entirely missed the R-R interval, while there was no difference in the HR monitor time stamp[a] | Artifact noted but not corrected |
| T6b | When the HR monitor entirely missed the R-R interval and there was a larger than expected gap in the HR monitor time stamp | The known interval subtracted from the difference in time stamp |

[a]Artifact is undetectable without simultaneous ECG recording.
[b]R-R interval of 0 ms was placed to maintain time synchrony between both devices.

The automatic correction (AC) and threshold-based correction (TBC) methods of Kubios HRV Premium (ver. 3.2) were used to automatically correct the artifacts. For the AC, "Automatic Correction" under the "R-R Interval Series Option" was selected and then the "Apply" button was selected to correct the artifacts. For the TBC, an appropriate threshold among very low (0.45 sec), low (0.35 sec), medium (0.25 sec), strong (0.15 sec), very strong (0.05 sec), and custom under "R-R Interval Series Option" was selected and then the "Apply" button was selected to correct the artifacts. When the artifacts were corrected with the TBC, the lowest level of correction was chosen to prevent potential overcorrection. When artifacts were detected by both methods, they were automatically replaced through the cubic spline interpolation by the software package. After the artifact corrections were made, the R-R interval time series was then considered normal and defined as N-N intervals [27].

## Calculation of HRV measures

Following the identification and correction of the artifacts, an identical segment of N-N intervals was selected from the ECG and Polar V800™ for the calculation of the various HRV domain measures. Commercial Kubios HRV Premium (ver. 3.2) analyzed the selected segments to obtain the time, frequency, and non-linear domains of HRV measures.

The time domain measures quantify the amount of variability within the sample and they included: standard deviation of normal R-R intervals (SDNN), root mean square of successive differences in normal R-R intervals (RMSSD), and percentage of successive normal R-R intervals greater than 50 ms (pNN50%). Frequency domain or power spectral density analysis reveal the content of signal's power (variance) versus frequency and can be analyzed by the autoregressive and fast Fourier transformation methods [27]. A fast Fourier transformation was performed to quantify power spectrum density into absolute low frequency (LF $ms^2$: 0.04–0.15 Hz) and absolute high frequency (HF $ms^2$: 0.15–0.40 Hz) domains. In addition, normalized LF (LF nu), normalized HF (HF nu), and LF/HF ratio were calculated. Non-linear domain measures represent the chaotic heartbeat dynamics caused by the complex interactions between several regulatory systems. We included sample entropy (SampEn) as a measure of non-linear methods. SampEn measures the signal complexity or irregularity where large SampleEn suggests high irregularity and smaller SampleEn a more regular signal [11].

## Measurement of R-wave amplitude

The amplitude of the R-wave from the subjects' ECG trace was measured in mm (10 mm = 1 mV) from the PR segment to the top of the R wave and averaged for each subject's lead [28]. Lead II was chosen for the R-wave amplitude measurement because it lies close to the cardiac axis (i.e., the overall direction of electrical movement), allowing for the best views of R-waves [29].

## Statistical analysis

The independent sample t-test was used to report the mean differences between the subject characteristics of men and women. The magnitude of difference between the R-R intervals and HRV domain measures from the ECG and Polar V800™ HR monitor were calculated by measuring the effect size (ES) as the mean difference over the standard deviation of the difference [30]. The ES was defined as trivial when ES <0.2, small when ES was between ≥0.2 and <0.5, moderate when ES between ≥0.5 and <0.8, and large when ≥0.8 [31]. The intraclass correlation coefficient (ICC) with the 95% confidence interval (CI) assessed the concurrent validity (or interchangeable agreement) of the R-R intervals and HRV measures [32]. We selected a two-way mixed model with absolute agreement and reported absolute measures (mean of two

measurements). The ICC was defined as poor when ICC was <0.50, moderate when ICC was between ≥0.50 and <0.75, good when ICC was between ≥0.75 and <0.90, and excellent when ICC was ≥0.90 [33]. Bland-Altman plots were created for the ECG R-R intervals versus the Polar V800™ HR monitor for the uncorrected (UN), AC, TBC, and MC R-R intervals. The 95% limits of agreement (LoA) for lower (-1.96) and upper limit (+1.96) were calculated as follows: 1) lower limit: mean difference—(SD of difference x 1.96); and 2) upper limit: (SD of difference x 1.96) + mean difference [34]. Homoscedasticity and heteroscedasticity were inspected through a histogram and Q-Q plot. The RMSSD, pNN50%, LFms², and HFms² were logarithmically transformed before the calculation of LoA ranges. Finally, the sub-group analyses of gender, BMI, medication use, and aerobic capacity were performed using the same statistical methods as in all subjects to study their potential impacts on the agreement between Polar V800™ and ECG. Gender and medication use were grouped into "male and female" and "medication and non-medication". Further, BMI and aerobic capacity were divided into "overweight (25.0–29.9 kg/m²)" and "obesity (>30 kg/m²)" and "very poor to poor (15.6–37.5 ml/kg/min)", "fair to good (32.0–41.5 ml/kg/min)", and "excellent (48.6–51.7 ml/kg/min)", respectively [25]. All statistical analyses were conducted using SPSS (ver. 24; Chicago, IL, USA).

## Results

### Subject characteristics

Participants were 17 men (68%) and 8 women (28%) between 18 and 55 years who were overweight to obese with hypertension and of very poor to good cardiorespiratory fitness for men and women of their age Table 2. Of these, six subjects were taking antihypertension medication that included diuretics (n = 1), angiotensin II receptor antagonists (n = 3), and β-Blockers (n = 2) S1 Dataset.

### Agreement of R-R intervals

The R-wave amplitude of ECG lead II was 0.26±0.09 mV, which represents a depressed R-wave amplitude defined as R-wave amplitude of 0.5 mV or less compared to healthy individuals (29). There were 71 artifacts among the 8325 total R-R intervals yielding an error rate of 0.85%. The most often detected type of error among the total number of artifacts included T4 (49x; 69%) followed by T5 (7x; 9.9%), T1 (5x; 7.0%), T3 (5x; 7.0%), T6a (2x; 2.8%), T6b (2x; 2.8%), and T2 (1x; 1.4%). The length of HRV measurement for eight subjects was less than 5 min due to the loss of connection between the HR monitor and the strap for unknown reasons.

**Table 2. Subjects characteristics of the total sample and by sex (mean ± SD).**

| Variable | Total Sample (n = 25) | Men (n = 17) | Women (n = 8) |
|---|---|---|---|
| Age (year) | 44.7±10.1 | 42.5±10.3 | 49.4±8.4 |
| Height (cm) | 172.3±11.0 | 176.9±9.6[a] | 162.6±5.4 |
| Weight (kg) | 93.7±30.6 | 90.9±16.2 | 86.4±17.2 |
| Body Mass Index (kg.m$^{-2}$) | 29.8±4.3 | 28.2±3.9 | 32.0±5.3 |
| Waist Circumference (cm) | 99.9±12.1 | 95.9±10.7 | 105.1±14.0 |
| Heart Rate (bpm) | 73.6±11.0 | 71.8±11.9 | 78.4±7.2 |
| Systolic Blood Pressure (mmHg) | 132.3±12.2 | 129.6±11.8 [a] | 140.9±7.9 |
| Diastolic Blood Pressure (mmHg) | 84.3±10.2 | 84.4±11.3 | 86.5±7.5 |
| Maximum Oxygen Consumption (VO$_{2peak}$) | 33.5±9.8 | 38.0±8.2[a] | 24.1±5.5 |

[a]Significant differences at $P < 0.05$.

Therefore, the average length of the HRV measurement was 4.6±0.9 min and number of R-R intervals was 333.0±70.5.

Fig 1 contains the Bland-Altman plots of the LoA and agreement between the R-R intervals from the ECG and the UN, AC, TBC, and MC R-R intervals from the Polar V800™ HR monitor. The UN R-R intervals from the Polar V800™ HR monitor were corrected using the AC and TBC methods of Kubios Premium (ver. 3.2) and MC. While the bias (0.69 ms) and ES (0.004) were small, the UN R-R intervals resulted in the widest range of LoA (from -215.80 to 214.42 ms) (Fig 1a). The AC method using Kubios HRV Premium (ver. 3.2) resulted in corrected R-R intervals with higher bias (3.79 ms), range of LoA (from -130.32 to 137.90 ms), and ES (0.024) (Fig 1b) than the bias (1.16 ms), range of LoA (from -92.67 to 94.98 ms), and ES (0.008) of the TBC method (Fig 1c). The MC method produced corrected R-R intervals with the smallest bias (0.37 ms), tightest range of LoA (-41.20 to 41.94 ms), and smallest effect size (0.002) (Fig 1d). Furthermore, an improvement in the ICC of the UN R-R intervals occurred depending on the method of correction that was used. The ICC of UN R-R intervals went from 0.79 (95%

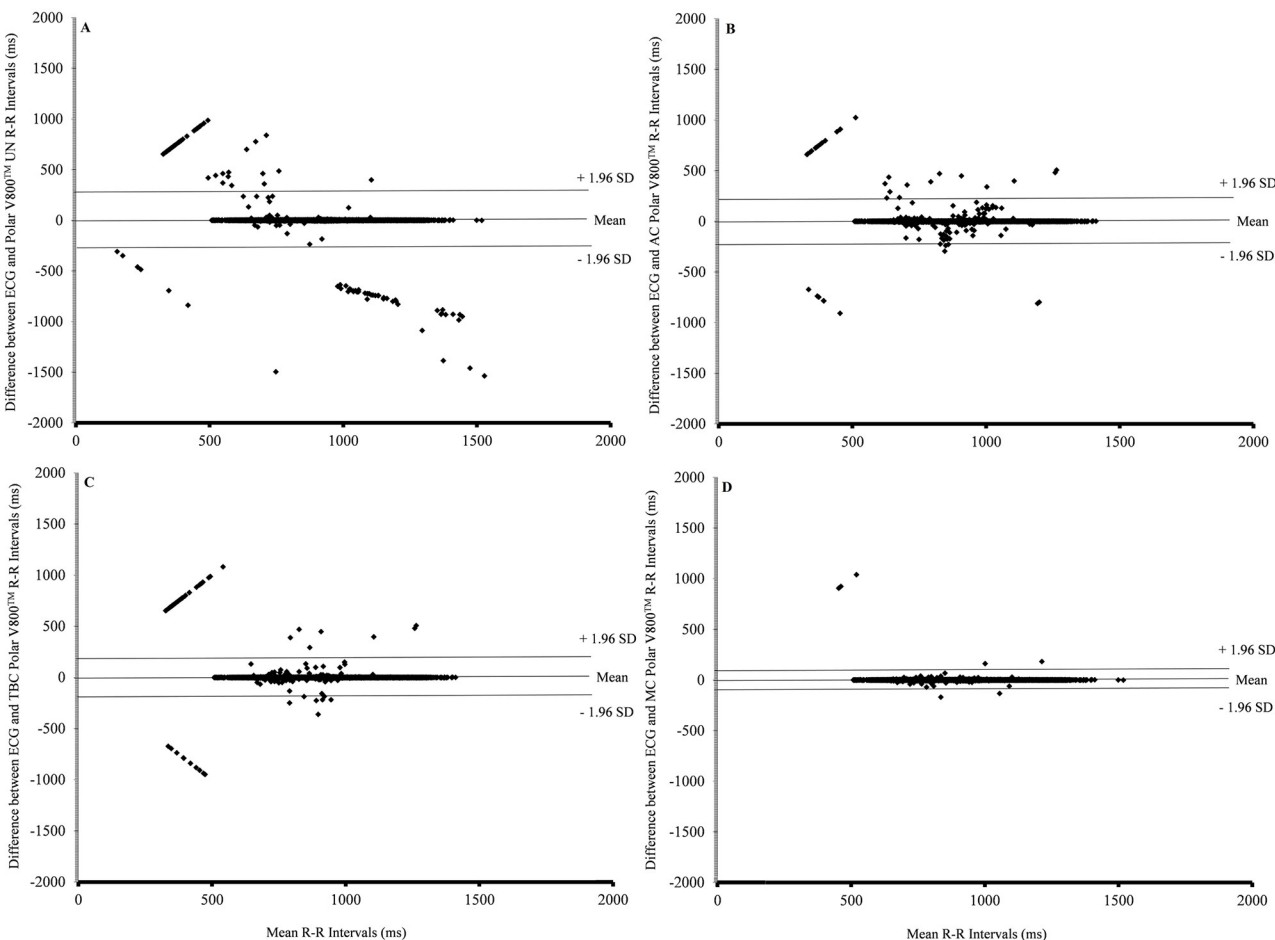

**Fig 1.** Bland-Altman plots for the ECG and the UN Polar V800™ HR monitor R-R intervals (a) Bland-Altman plots for the ECG and the AC Polar V800™ HR monitor R-R intervals (b) Bland-Altman plots for the ECG and the TBC Polar V800™ HR monitor R-R intervals (c) Bland-Altman plots for the ECG and the MC Polar V800™ HR monitor R-R intervals (d) ECG: electrocardiogram; HR: heart rate; UN: uncorrected R-R intervals; AC: R-R intervals corrected by automatic correction method of the Kubios HRV Premium (ver 3.2); TBC: R-R intervals corrected by threshold-based correction method of the Kubios HRV Premium (ver 3.2); MC: R-R intervals corrected manually.

Table 3. Comparison of HRV measures calculated from UN Polar V800™ and ECG R-R intervals (mean ± SD).

| HRV Measure | ECG (mean±SD) | Polar UN (mean±SD) | Bias (LoA) | ICC (95% CI) | Effect Size |
|---|---|---|---|---|---|
| SDNN (ms) | 55.5±26.7 | 90.5±62.2 | -34.95 (-143.58 to 73.69) | 0.27 (-0.07–0.57) | 0.730 |
| RMSSD (ms) | 38.3±29.4 | 92.8±96.3 | -57.51 (-223.81 to 114.78) | 0.21 (-0.11–0.52) | 0.765 |
| pNN50 (%) | 13.6±16.2 | 15.1±15.9 | -1.50 (-6.14 to 3.21) | 0.98 (0.95–0.99) | 0.091 |
| LF (ms$^2$) | 1215±1352.1 | 6924.2±16947.2 | -5709.13 (-38536.30 to 27118.05) | 0.53 (-0.97–0.57) | 0.475 |
| HF (ms$^2$) | 740.8±1262.8 | 4861.0±6863.9 | -4120.18 (-17136.40 to 8896.033) | 0.07 (-0.21–0.39) | 0.835 |
| LF (nu) | 67.3±18.7 | 59.7±20.3 | 7.53 (-30.13 to 45.20) | 0.49 (0.14–0.73) | 0.387 |
| HF (nu) | 32.7±18.6 | 40.2±20.2 | -7.53 (-45.20 to 30.13) | 0.49 (0.14–0.73) | 0.387 |
| LF/HF Ratio | 3.6±4.0 | 3.0±4.2 | 0.67 (-3.37 to 4.71) | 0.87 (0.72–0.94) | 0.163 |
| SampEn | 1.5±0.2 | 1.2±0.5 | 0.33± (-0.58 to 1.24) | 0.39 (-0.18 to 0.71) | 0.836 |

ECG: electrocardiogram; UN: uncorrected R-R intervals; ICC: intra-correlation coefficient; LoA: limits of agreement; SDNN: standard deviation of normal-to-normal N-N intervals; RMSSD: root mean square of the successive difference of intervals; pNN50%: the percentage of successive normal cardiac inter-beat intervals greater than 50 msec; LF: low frequency; HF: high frequency; SampEn: Sample Entropy; SD: standard deviation.

CI 0.78–0.80) to 0.91 (95% CI 0.90–0.91) in the AC R-R intervals, 0.95 (95% CI 0.95–0.95) in the TBC R-R intervals, and 1.00 (95% CI 1.00–1.00) in MC R-R intervals S2 Dataset.

## Agreement of HRV measures

Tables 3, 4, 5 and 6 contain the comparisons of the HRV measures calculated from UN, AC, TBC, and MC Polar V800™ HR monitor R-R intervals versus those calculated from ECG R-R intervals using the Kubios Premium (ver. 3.2) software.

The UN resulted in the largest biases and the widest LoA ranges for all HRV measures compared to the AC, TBC, and MC methods, except for pNN50% (ES: 0.091; ICC: 0.98) and LF/HF ratio (ES: 0.163; ICC: 0.87) Table 3. The UN ES ranged from 0.091 to 0.835, indicating the magnitude of difference between the HRV measures calculated from UN Polar V800™ HR monitor and ECG R-R intervals was trivial to large, and had five out of 9 HRV domain measures between $0.2 \leq$ and $<0.5$, indicating a moderate difference. Additionally, the UN ICCs ranged from 0.070 to 0.98 suggesting poor to excellent agreement, but had six out of nine HRV measures below $<0.5$, indicating poor agreement. Between the two correction methods of Kubios HRV Premium (ver. 3.2), the TBC resulted in smaller biases than the AC for SDNN, RMSSD, pNN50%, LFnu, HFnu, LF/HF ratio, but the AC produced smaller biases than TBC for LFms$^2$, HFms$^2$ and SampEn Tables 4 and 5. Additionally, when the TBC was compared to

Table 4. Comparison of HRV measures calculated from Kubios Premium (ver. 3.2) AC Polar V800™ and ECG R-R intervals (mean ± SD).

| HRV Measure | ECG (mean±SD) | Polar AC (mean±SD) | Bias (LoA) | ICC (95% CI) | Effect Size |
|---|---|---|---|---|---|
| SDNN (ms) | 55.5±26.7 | 57.1±28.1 | -1.60 (-23.27 to 20.06) | 0.92 (0.83–0.96) | 0.058 |
| RMSSD (ms) | 38.3±29.4 | 36.2±25.4 | 2.13 (-25.57 to 29.84) | 0.87 (0.73–0.94) | 0.078 |
| pNN50 (%) | 13.6±16.2 | 13.1±16.1 | 0.51 (-7.28 to 8.29) | 0.97 (0.93–0.99) | 0.031 |
| LF (ms$^2$) | 1215±1352.1 | 1214.2±1360.9 | 0.80 (-369.45 to 371.06) | 0.99 (0.98–1.00) | 0.001 |
| HF (ms$^2$) | 740.8±1262.8 | 749.1±1239.5 | -8.27 (-1115.84 to 1099.293) | 0.90 (0.79–0.96) | 0.007 |
| LF (nu) | 67.3±18.7 | 66.7±19.6 | 0.57 (-24.29 to 25.44) | 0.79 (0.57–0.90) | 0.030 |
| HF (nu) | 32.7±18.6 | 33.3±19.6 | -0.57 (-25.44 to 24.29) | 0.79 (0.73–0.95) | 0.030 |
| LF/HF Ratio | 3.6±4.0 | 3.5±3.9 | 0.12 (-1.76 to 2.00) | 0.97 (0.94–0.99) | 0.031 |
| SampEn | 1.5±0.2 | 1.5±0.3 | 0.02 (-0.21 to 0.21) | 0.81 (0.57–0.91) | 0.039 |

AC: R-R intervals corrected by automatic correction method of Kubios HRV Premium (ver. 3.2).

**Table 5. Comparison of HRV measures calculated from Kubios Premium (ver. 3.2) TBC Polar V800™ and ECG R-R intervals (mean ± SD).**

| HRV Measure | ECG (mean±SD) | Polar TBC (mean±SD) | Bias (LoA) | ICC (95% CI) | Effect Size |
|---|---|---|---|---|---|
| SDNN (ms) | 55.5±26.7 | 56.1±27.6 | -0.55 (-10.59 to 9.49) | 0.98 (0.96–0.99) | 0.020 |
| RMSSD (ms) | 38.3±29.4 | 37.8±27.5 | 0.53 (-9.57 to 10.63) | 0.98 (0.96–0.99) | 0.019 |
| pNN50 (%) | 13.6±16.2 | 13.8±16.2 | -0.16 (-1.33 to 1.02) | 0.99 (0.98–1.00) | 0.010 |
| LF (ms$^2$) | 1215±1352.1 | 1209.2±1352.5 | 5.86 (-275.86 to 287.59) | 1.00 (0.99–1.00) | 0.004 |
| HF (ms$^2$) | 740.8±1262.8 | 703.7±1201.6 | 37.07 (-460.59 to 534.74) | 0.98 (0.95–0.99) | 0.030 |
| LF (nu) | 67.3±18.7 | 67.3±18.0 | -0.02 (-4.04 to 4.01) | 0.99 (0.99–1.00) | 0.001 |
| HF (nu) | 32.7±18.6 | 32.7±18.0 | 0.02 (-4.01 to 4.04) | 0.99 (0.99–1.00) | 0.001 |
| LF/HF Ratio | 3.6±4.0 | 3.5±3.9 | 0.12 (-0.79 to 1.02) | 0.99 (0.98–1.00) | 0.029 |
| SampEn | 1.5±0.2 | 1.5±0.3 | 0.02 (-0.35 to 0.40) | 0.93 (0.85–0.97) | 0.039 |

TBC: R-R intervals corrected by threshold-based correction method of Kubios HRV Premium (ver. 3.2).

the AC method, the TBC produced tighter LoA ranges and smaller ES (<0.031 versus <0.085) for all HRV domain measures except for SampEn.

Moreover, the TBC method exhibited ICCs varying from 0.96 to 1.00, while the AC had ICCs ranging from 0.79 to 0.99, indicating that the agreeability slightly improved from good-to-excellent to excellent when the correction was made with the TBC rather than AC. The MC produced the smallest biases for SDNN, RMSSD, pNN50%, LF/HF, and SampEn compared to the AC and TBC methods but resulted in larger biases for LFms$^2$ and HFms$^2$ than AC, and for LFms$^2$, LFnu, and HFnu than TBC Table 6. Additionally, when the MC was compared to the AC and TBC methods, the MC resulted in the tightest LoA ranges and smallest ES (<0.012) for all HRV measures. Furthermore, the MC had ICC of 0.99 or 1.00 for all HRV measures indicating excellent agreement for all HRV measures S3 Dataset.

## Agreement of HRV measures separated by medication use, gender, BMI, and aerobic capacity

The magnitude of biases, LoA ranges, ICCs, and effect sizes of HRV measures of the subjects with medication use S2A–S2D Table, males S2A–S2D Table, those with overweight BMI S3A–S3D Table, and those with very poor-to-poor aerobic capacity S4A–S4D Table were similar to those of non-medication, female, obesity, and excellent aerobic capacity between the Polar V800™ and ECG. Also, as was in all subjects, the correction of artifacts substantially improved

**Table 6. Comparison of HRV measures calculated from MC R-R intervals between the ECG and polar (mean ± SD).**

| HRV Measure | ECG (mean±SD) | Polar MC (mean±SD) | Bias (LoA) | ICC (95% CI) | Effect Size |
|---|---|---|---|---|---|
| SDNN (ms) | 55.5±26.7 | 55.3±26.7 | 0.21 (-0.70 to 1.12) | 1.00 (1.00–1.00) | 0.009 |
| RMSSD (ms) | 38.3±29.4 | 38.1±29.1 | 0.18 (-1.82 to 2.18) | 1.00 (0.99–1.00) | 0.006 |
| pNN50 (%) | 13.6±16.2 | 13.7±16.1 | -0.11 (-1.52 to 1.30) | 1.00 (0.99–1.00) | 0.007 |
| LF (ms$^2$) | 1215±1352.1 | 1208.2±1351 | 6.85 (-65.77 to 79.48) | 1.00 (1.00–1.00) | 0.005 |
| HF (ms$^2$) | 740.8±1262.8 | 730.9±1267.1 | 9.94 (-44.20 to 64.07) | 1.00 (1.00–1.00) | 0.008 |
| LF (nu) | 67.3±18.7 | 67.6±18.6 | -0.35 (-1.68 to 0.98) | 1.00 (1.00–1.00) | 0.019 |
| HF (nu) | 32.7±18.6 | 32.3±18.6 | 0.35 (-0.98 to 1.68) | 1.00 (1.00–1.00) | 0.019 |
| LF/HF Ratio | 3.6±4.0 | 3.7±4.0 | -0.04 (-0.23 to 0.14) | 1.00 (1.00–1.00) | 0.011 |
| SampEn | 1.5±0.2 | 1.5±0.2 | 0.01 (-0.11 to 0.12) | 0.97 (0.93–0.99) | 0.001 |

MC: R-R intervals corrected manually.

the disagreement seen between UN HRV measures of the Polar V800™ and ECG. Moreover, the accuracy of artifact correction methods in correcting R-R intervals were similar to those reported in all subjects (AC<TBC≤AC) S4–S7 Datasets.

## Discussion

We compared uncorrected and corrected R-R intervals and HRV measures derived from the Polar V800™ HR monitor to the gold standard 12-lead ECG, with the aim of determining their level of agreement among adults with hypertension. In addition, we sought to determine the accuracy of the MC, AC and TBC methods of Kubios HRV Premium (ver.3.2) in correcting artifacts among this sample. Artifact correction was required due to the large biases, LoA ranges, ES, and poor ICC of HRV measures between the ECG and uncorrected Polar V800™ HR monitor data with the exceptions of pNN50% and LF/HF ratio. The results demonstrate that the MC consistently produced the lowest biases, LoA ranges, ES, and highest ICC for all correction methods followed by TBC and AC in turn. The results showed our subjects had a depressed R-wave amplitude that might have resulted from the clustering effect of being over-weight to obese and having hypertension [8, 29], increasing the likelihood of the occurrence of artifacts in this population.

The rate of the occurrence of artifacts in the Polar V800 R-R interval time series in the current study population (0.85%) was higher than those reported by Giles et al. [22] (0.08%) and [23] (0.10%) who originally validated the Polar V800™ among healthy adults (n = 20 [3 women and 17 men]; n = 18 [all men]) with a normal BMI (18.5–24.9 kg.m$^{-2}$). The data acquisition and analyses methods were the same between our study and the other two studies. This suggests that the performance of the Polar V800™ in the detection of R-R intervals somewhat declined in adults with hypertension. Typically, a low R-wave amplitude is associated with overweight or obesity [29] due to the electrical insulating role of excess epicardial and subcutaneous adipose tissue on chest wall of individuals with these conditions [9]. Additionally, Bacharova et al. [8] found an association between low R-wave amplitude among adults in the early stages of hypertension without structural changes in myocardium. Therefore, the low R-wave amplitude of 0.26±0.09 mV we found in the subjects in our study could be the result of them being overweight to obese, having hypertension, or both. This may also explain the higher occurrence of technical artifacts in this study compared to those reported by Giles et al. [22] and [23] among healthy adults with normal weight and BP as diminished R-wave amplitude increases the chance of artifact occurrence [7].

We identified the artifacts in the Polar V800™ R-R interval time series following the guidelines established by Giles et al. [22]. The most common type of error we encountered with the Polar V800™ uncorrected R-R intervals was T4 of 69%, which is often attributed to a decrease in R-wave amplitude due to a loss or decrease in connection between the chest strap and skin [22]. However, the depression of R-wave amplitude due to the subjects in our study being overweight to obese and having hypertension may have contributed to the greater occurrence of T4 errors compared to other studies involving healthy subjects [18, 22, 23]. Of note, the Polar V800TM uncorrected T1 and T6a errors can be seen on the corrected Bland–Altman plots (Fig 1D) as outliers, a similar finding reported by Giles et al. [22]. T1 and T6 errors are not visible without simultaneous ECG recording.

The uncorrected R-R intervals resulted in large biases and LoA ranges, poor ICC, and large ES, necessitating the correction of artifacts prior to calculation of HRV measures, as is recommended [27]. The MC was the most valid method of correction with the lowest bias, tightest LoA ranges, and highest ICC compared to the AC and TBC methods. Time, frequency, and non-linear HRV measures calculated from the Polar V800™ R-R intervals corrected by the MC

and those calculated from ECG showed excellent agreement with consistent small biases, tight LoA ranges, trivial ES, and excellent ICC, similar to levels of agreement found in previous studies with the Polar V800™ [22–24].

The MC of artifacts in R-R interval time series is time consuming, thus automated tools have been developed for the identification and correction of such data. Kubios HRV Premium is possibly the most extensively used software package for this purpose. However, we observed the TBC of Kubios HRV Premium (ver. 3.2) was not able to properly correct T4 error containing $\geq$2 missed beats, the same issue detected previously by [23] with Kubios HRV (ver. 2.2). More specifically, the TBC replaced the T4 error containing $\geq$2 missed beats with one interval rather than the required $\geq$2, leading to HRV measures with wider LoA ranges than the MC. Therefore, correcting artifacts with the TBC may be the more convenient option for users who find the MC complicated and laborious.

The AC of Kubios HRV was the least accurate correction method with the largest biases, LoA ranges, ES and lowest ICCs compared to the MC and TBC, which may be explained by two critical issues we identified with the AC method. First, the AC was not able to properly correct T4 errors containing >2 missed beats, rather the erroneous beat was replaced with one interval rather than the required >2 intervals. Second, the AC always overcorrected three intervals at the beginning and end of an R-R interval time series whether there were artifacts or not. Considering even a single distortion in R-R intervals results in exaggerated measures of HRV [6, 35, 36] overcorrecting six intervals despite no artifacts present may be largely the reason for the poorest performance of the AC compared to the MC and TBC.

This is the first study to validate the Polar V800™ HR monitor in detecting R-R intervals and producing agreeable HRV measures among adults with hypertension who were also overweight or obese. The occurrence of technical artifacts was greater in our sample than in healthy individuals with normal weight and BP, necessitating the need for artifact correction in this clinical population. General knowledge and prior experience of HRV data and artifact correction are essential to ensure optimum data quality regardless of the selection of artifact correction method. Ideally, users should choose MC as a first option for correcting Polar V800™ artifacts following the current artifact correction guidelines, as MC results in the tightest LoA ranges, smallest ES, and excellent ICC compared to the automatic correction methods of Kubios HRV Premium (ver. 3.2.). However, Kubios HRV Premium (ver. 3.2.) methods may be of greater practical utility for high throughput situations where the correction of artifacts using MC is not practical. In that case, the TBC should be selected as the preferred option over AC since TBC produced smaller biases than the AC for all HRV measures except LF ms$^2$ and HFms$^2$ as well as tighter LoAs, and smaller ES for all HRV measures. Therefore, if MC is not feasible, TBC should be the method of choice when choosing a method of correction for Polar V800™ artifacts among adults with hypertension. Finally, Kubios should improve its automatic correction methods by addressing the previously mentioned issues with TBC and AC to optimize HRV data quality among this population.

## Supporting information

**S1 Dataset. Individual subject characteristics.**
(XLSX)

**S2 Dataset. Individual UN, AC, TBC, and MC R-R intervals.**
(XLSX)

**S3 Dataset. Individual HRV measures calculated from the UN, AC, TBC, and MC R-R intervals.**
(PDF)

**S4 Dataset. Individual HRV measures separated by medication use calculated from UN, AC, TBC, and MC R-R intervals.**
(XLSX)

**S5 Dataset. Individual HRV measures separated by gender calculated from UN, AC, TBC, and MC R-R intervals.**
(XLSX)

**S6 Dataset. Individual HRV measures separated by BMI calculated from UN, AC, TBC, and MC R-R intervals.**
(XLSX)

**S7 Dataset. Individual HRV measures separated by aerobic capacity calculated from UN, AC, TBC, and MC R-R intervals.**
(XLSX)

**S1 Table. (A-D) Comparison of HRV measures separated by medication use calculated from UN, AC, TBC, and MC R-R intervals (mean ± SD).**
(PDF)

**S2 Table. (A-D) Comparison of HRV measures separated by gender calculated from UN, AC, TBC, and MC R-R intervals (mean ± SD).**
(PDF)

**S3 Table. (A-D) Comparison of HRV measures separated by BMI calculated from UN, AC, TBC, and MC R-R intervals (mean ± SD).**
(PDF)

**S4 Table. (A-D) Comparison of HRV measures separated by aerobic capacity calculated from UN, AC, TBC, and MC R-R intervals (mean ± SD).**
(PDF)

## Acknowledgments

The authors acknowledge the research assistance provided by the PULSE and FITT and FIRED UP investigative team and the enthusiastic support from our participants and their families.

## Author Contributions

**Conceptualization:** Burak Cilhoroz, Amanda Zaleski, Beth Taylor.

**Data curation:** Burak Cilhoroz, David Giles.

**Formal analysis:** Burak Cilhoroz.

**Funding acquisition:** Linda Pescatello.

**Investigation:** Burak Cilhoroz, Amanda Zaleski, Beth Taylor, Bo Fernhall, Linda Pescatello.

**Methodology:** Burak Cilhoroz, David Giles, Amanda Zaleski, Bo Fernhall, Linda Pescatello.

**Project administration:** Burak Cilhoroz, Linda Pescatello.

**Resources:** Linda Pescatello.

**Software:** Burak Cilhoroz.

**Supervision:** Amanda Zaleski, Beth Taylor, Bo Fernhall, Linda Pescatello.

**Validation:** Burak Cilhoroz, Linda Pescatello.

**Visualization:** Burak Cilhoroz, David Giles, Linda Pescatello.

**Writing – original draft:** Burak Cilhoroz.

**Writing – review & editing:** Burak Cilhoroz, David Giles, Amanda Zaleski, Beth Taylor, Bo Fernhall, Linda Pescatello.

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
