## [Decision Letter · Decision Letter 0]

23 Jul 2020

PONE-D-20-14879

Validation of the Polar V800TM Heart Rate Monitor and Comparison of Artifact Correction Methods among Adults with Hypertension

PLOS ONE

Dear Dr. Cilhoroz,

Thank you for submitting your manuscript to PLOS ONE. After careful consideration, we feel that it has merit but does not fully meet PLOS ONE’s publication criteria as it currently stands. Therefore, we invite you to submit a revised version of the manuscript that addresses the points raised during the review process.

We look forward to receiving your revised manuscript.

Kind regards,

Daniel Boullosa

Academic Editor

PLOS ONE

Journal Requirements:

Reviewers' comments:

Reviewer's Responses to Questions

**Comments to the Author**

1. Is the manuscript technically sound, and do the data support the conclusions?

Reviewer #1: No

2. Has the statistical analysis been performed appropriately and rigorously? 

Reviewer #1: No

3. Have the authors made all data underlying the findings in their manuscript fully available?

Reviewer #1: No

4. Is the manuscript presented in an intelligible fashion and written in standard English?

Reviewer #1: Yes

5. Review Comments to the Author

Reviewer #1: Review of manuscript PONE-D-20-14879 entitled "Validation of the Polar V800TM Heart Rate Monitor and Comparison of Artifact Correction Methods among Adults with Hypertension" for the PLOS ONE.

The Polar V800 sport tester have been verified in a group of healthy subjects and athletes. This manuscript validated the Polar V800 for a group of overweight and obese subjects. I still lack such a study in the literature. The results of this manuscript are important because it is known that overweight and obese subjects have poor heart rate variability (HRV) compared to healthy subjects. Therefore, HRV of overweight and obese subjects should be monitored, preferably at home. Small, lightweight and cheap devices for ECG recording, such as a sport tester, enable the development of telemedicine. In addition, this manuscript compared several methods of artifact detection and correction. In my opinion, this important issue is not sufficiently discussed in the literature. I therefore consider this manuscript to be an interesting and it adds knowledge in the literature. However, there are some issues in the manuscript that should be solved. Please see my comments below.

Page 5: You have stated that all relevant data are within the manuscript and its Supporting Information files. However, I cannot find any data files from which the results are based.

Page 8: I propose to replace “… beyond that of healthy populations.” with “… beyond that of patients”. The use of medical ECG device has usually been limited to the diagnostics of patients with suspected cardiovascular condition.

Page 9: You should mention the negative factors that are associated with hypertension – excessive body mass and/or low physical activity.

Page 9: You should describe what HRV is in a few sentences.

Page 9: I propose to replace “… with technical (e.g. missed beats) …” with “… with technical (e.g. excessive noise which causes incorrect detection of R peak) ...”

Page 9: I propose to replace “… and physiological (e.g. non-sinus beats) …” with “… and physiological (e.g. ectopic beat) …”

Page 10: I do not think atherosclerosis is the result of poor ANS activity. This is more due to a change in distribution of different types of cholesterol and hormonal imbalances.

Page 13: “breaths/min–1” is incorrect. Please replace with “breaths⋅min–1” or “breaths/min”.

Page 14: Is peak oxygen consumption really a potential covariate of sport tester validity? How was it tested in this manuscript? I have found very limited information in the result section.

Page 14: The subtitle “R-R Interval Recording” is rather “R-R Interval Processing”. Please consider rephrasing.

Page 14: “BTC” should be replaced by “one of the authors (BTC)”.

Page 14: So, the correct position of the R peak was performed by only one author? Given that R-R intervals obtained from a medical ECG device were used as a gold standard in this manuscript, I consider that a single author checking is not sufficient.

Page 14: Please describe the MC method in more detail. Who performed it? In which kind of software? Did the person have RR intervals available from the medical ECG device or not?

Page 16: “… in both signals (N = 3)” does it mean three occurrences of artifacts in all group of RR recordings? Or does it mean that recordings from three subjects contains artifacts?

Page 17: “unpredictable heartbeat dynamics” should be replaced by ”chaotic heartbeat dynamics”. A chaotic system is theorethically predictable, but it is a complex nonlinear system, so in practise the chaotic system looks stochastic (random). Mathematical theory dealing with nonlinear HRV indexes considers ANS to be chaotic, not stochastic. See, for example, Ernst G (2014). Heart rate variability. London: Springer.

Page 17: I agree that Poincaré plot is one of the graphical methods used to study chaotic systems. Poincaré plot can therefore be considered as a nonlinear method. However, I cannot agree that SD1, SD2, and SD2/SD1 can be considered as nonlinear indexes of HRV. It was shown that these indexes can be calculated from SDNN and SDSD (Sassi R, Cerutti S, Lombardi F, Malik M, Huikuri H V, Peng C-K, … Yamamoto Y (2015). Advances in heart rate variability signal analysis: Joint position statement by the e-cardiology ESC working group and the european heart rhythm association co-endorsed by the asia pacific heart rhythm society. Europace, 17, 1341–1353). Therefore these indexes should be considered as time domain indexes. Sample entropy is nonlinear HRV index suitable for short-term (≈5 min) RR recordings. Therefore, I suggest removing the indexes SD1, SD2, and SD1/SD2 and adding sample entropy.

Page 18: ”inter correlation coeficient” should be replaced by “intraclass correlation coefficient”.

Page 18: What type of ICC was used? There are several types of ICC. Please specify the ICC you used according to Koo & Li (2016) or a similar source.

Page 18: You did not specify which variables were treated by the logarithmic transformation. Please add this information.

Page 18/19: “The main findings of gender, BMI, medication use, and aerobic capacity sub-groups were similar to those in all subjects. Moreover, the removal of subjects using B-blockers from the analyses did not have an impact on the findings.” Please be more specific or remove these sentences.

Page 19, Table 2: There is a significant difference in systolic blood pressure. However, no statistical procedure is described in the statistical analysis subsection. Is it a two-sample t-test?

Page 19: What does “Agreeability and interchangeability” mean? I suggest to use the term agreement.

Page 19: Why is the average length of the HRV measurement 4.6±0.9 min? I expect a value of 5 min.

Page 19, Table 6: The results of 1.00 (1.00 – 1.00) for ICC (95% CI) look strange because ICC = 1 means a perfect agreement that cannot be reached in practise. In your manuscript, the ICC could not be equal to 1 because Figure 1D shows several discrepancies and the associated LoAs are not zero. Maybe it is due to rounding. Please try to write the ICC value for more decimal digits if the ICC values I close to 1.

6. PLOS authors have the option to publish the peer review history of their article (what does this mean?). If published, this will include your full peer review and any attached files.

Reviewer #1: **Yes: **Jakub Krejci

---

## [Author Response · Author response to Decision Letter 0]

3 Sep 2020

Reviewers’ comments to Authors:

Reviewer #1: 

Comments to the Author:

The Polar V800 sport tester have been verified in a group of healthy subjects and athletes. This manuscript validated the Polar V800 for a group of overweight and obese subjects. I still lack such a study in the literature. The results of this manuscript are important because it is known that overweight and obese subjects have poor heart rate variability (HRV) compared to healthy subjects. Therefore, HRV of overweight and obese subjects should be monitored, preferably at home. Small, lightweight and cheap devices for ECG recording, such as a sport tester, enable the development of telemedicine. In addition, this manuscript compared several methods of artifact detection and correction. In my opinion, this important issue is not sufficiently discussed in the literature. I therefore consider this manuscript to be an interesting and it adds knowledge in the literature. However, there are some issues in the manuscript that should be solved. Please see my comments below.

Thank you for your supportive comment!

Page 5: You have stated that all relevant data are within the manuscript and its Supporting Information files. However, I cannot find any data files from which the results are based.

We thank the reviewer for their input about the missing supporting data files. All supporting information including individual subject characteristics, HRV measures, R-R intervals used to create figures as well as individual HRV measures separated by medication use, gender, BMI, and aerobic capacity (to show the potential influence of medication use, gender, BMI, and VO2peak on the agreement between ECG and Polar V800TM) were added as supporting information with following names:

Supporting Information

S1 Dataset. Individual Subject Characteristics.

S2 Dataset. Individual UN, AC, TBC, and MC R-R intervals.

S3 Dataset. Individual HRV measures calculated from the UN, AC, TBC, and MC R-R intervals.

S1A-S1D Tables. Comparison of HRV measures separated by medication use calculated from UN, AC, TBC, and MC R-R intervals (mean ± SD). 

S2A-S2D Tables. Comparison of HRV measures separated by gender calculated from UN, AC, TBC, and MC R-R intervals (mean ± SD). 

S3A-S3D Tables. Comparison of HRV measures separated by BMI calculated from UN, AC, TBC, and MC R-R intervals (mean ± SD). 

S4A-S4D Tables. Comparison of HRV measures separated by aerobic capacity calculated from UN, AC, TBC, and MC R-R intervals (mean ± SD). 

S4 Dataset. Individual HRV measures separated by medication use calculated from UN, AC, TBC, and MC R-R intervals. 

S5 Dataset. Individual HRV measures separated by gender calculated from UN, AC, TBC, and MC R-R intervals. 

S6 Dataset. Individual HRV measures separated by BMI calculated from UN, AC, TBC, and MC R-R intervals. 

S7 Dataset. Individual HRV measures separated by aerobic capacity calculated from UN, AC, TBC, and MC R-R intervals. 

Please note that the findings of subgroup analyses of HRV measures separated by medication use, gender, BMI, and aerobic capacity (S1A-S1A Tables; S2A-S2D Tables; S3A-S3D Tables, and S4A-S4D Tables) were added as supporting files to justify our following statement in page 18/19 “The main findings of gender, BMI, medication use, and aerobic capacity sub-groups were similar to those in all subjects. Moreover, the removal of subjects using B-blockers from the analyses did not have an impact on the findings.” Therefore, we also added the individual HRV measures of sub-group analyses as supporting information files (S4 Dataset; S5 Dataset; S6 Dataset; and S7 Dataset).

We have also cited our supplemental information in our manuscript text as “S1 Dataset” under “Subject Characteristics”, “S2 Dataset” under “Agreement of R-R Intervals”, “S3 Dataset” under “Agreement of HRV Measures”, S1A-S1A Tables; S2A-S2D Tables; S3A-S3D Tables, and S4A-S4D Tables” as well as “S4-S7 Datasets” under “Agreement of HRV Measures Separated by Medication Use, Gender, and Aerobic Capacity”

Page 8: I propose to replace “… beyond that of healthy populations.” with “… beyond that of patients”. The use of medical ECG device has usually been limited to the diagnostics of patients with suspected cardiovascular condition.

We acknowledge the confusing statement about the use of heart rate monitors in the first sentence of the abstract. Therefore, we have revised the first sentence of the abstract which originally stated:

“The apparent ease of measuring heart rate variability (HRV) via ambulatory monitors has extended their use beyond that of healthy populations.”

Which now reads as:

“Heart rate variability (HRV) measurements via ambulatory monitors have become common.”

Page 9: You should mention the negative factors that are associated with hypertension – excessive body mass and/or low physical activity.

We have made the following changes to state the risk factors associated with hypertension after the first sentence of introduction:

1) Added (second sentence): “Modifiable (e.g., obesity, physical inactivity, smoking, dyslipidemia) and non-modifiable (e.g., age, gender, family history) risk factors can increase the likelihood of developing hypertension [2].”

2) Changed: “The underlying etiology for the development of hypertension is largely unknown. One of the hypotheses regarding the initiation, progression, and maintenance of hypertension is alterations in the neural control of blood pressure (BP) [3].”

to 

“However, the underlying etiology for the development of hypertension is largely unknown. One of the hypotheses regarding the initiation, progression, and maintenance of hypertension is alterations in the neural control of blood pressure (BP) [3].”

Page 9: You should describe what HRV is in a few sentences.

We have made the following changes to describe HRV after the fourth sentence of the first paragraph of introduction:

1) Added (fifth sentence): “The sinoatrial node integrates the inputs from the ANS to adjust heart rate (HR) in response to the constantly changing internal and external environment to maintain homeostasis [4]).”

2) Added (sixth sentence): “This adjustment causes an oscillatory pattern in the HR resulting from beat-to-beat fluctuations in the time period between sequential heartbeats, termed heart rate variability (HRV).”

3) Changed (last sentence of the first paragraph of introduction): “Heart rate variability (HRV) is used to non-invasively measure ANS modulation.”

to 

“HRV is used to non-invasively measure ANS modulation.”

Page 9: I propose to replace “… with technical (e.g. missed beats) …” with “… with technical (e.g. excessive noise which causes incorrect detection of R peak) ...”

As per the reviewer’s request, we have made the following changes in third sentence of the second paragraph of introduction in page 9:

1) Deleted: “(e.g., missed beats)”

2) Added: “(e.g., excessive noise which causes incorrect detection of R peak)”

Page 9: I propose to replace “… and physiological (e.g. non-sinus beats) …” with “… and physiological (e.g. ectopic beat) …”

As per the reviewer’s request, we have made the following changes in third sentence of the second paragraph of introduction in page 9:

1) Deleted “(e.g., non-sinus beats)”

2) Added: “(e.g., ectopic beat)”

Page 10: I do not think atherosclerosis is the result of poor ANS activity. This is more due to a change in distribution of different types of cholesterol and hormonal imbalances.

We thank the reviewer for this insightful comment. Though we agree the roles of the changes in the distribution of different types of cholesterol and hormonal balances in the formation of atherosclerosis we also would like to point out different physiological pathways shown in literature to contribute to the development of atherosclerosis. Atherosclerosis is classified as an inflammatory disease, showing that inflammation is the crucial component of this condition (Libby P. History of discovery: inflammation in atherosclerosis. Arterioscler Thromb Vasc Biol2012;32:2045–2051). The literature suggests that increased expression of inflammation does not occur as a standalone condition but also the result of sympathetic activation. In fact, El-Armoche et al. (2009) argue that during the stress response (physical or psychological) the sympathetic system stimulates myocytes through the beta-adrenergic signaling system resulting in the activation of proinflammatory cytokines (e.g., interleukine 18). These authors, therefore, infer that chronic stimulation of the beta-adrenergic system triggered by excess sympathetic activation afflicts detrimental cardiotoxic impact on endothelial function, contributing to the development and progression of atherosclerosis (El-Armouche A , Eschenhagen T. β-Adrenergic stimulation and myocardial function in the failing heart. Heart Fail Rev2009;14:225–241). Moreover, Pizzi et al. (2010) and Ulleryd et al. (2017) contend that ANS dysfunction can be the cause of alterations in endothelial functions in which the increased level of proinflammatory cytokines may be the mediators of the effect of ANS dysfunction on atherosclerosis formation and progression (Pizzi C, Manzoli L, Mancini S, Bedetti G, Fontana F, Costa GM Atherosclerosis. 2010 Sep; 212(1):292-8; Ulleryd MA, Prahl U, Börsbo J, Schmidt C, Nilsson S, Bergström G, Johansson ME). 

Consequently, we retain stating the potential link between atherosclerosis and excess sympathetic activation. To make our statement clearer for the readers, however, we have made the following change in the second sentence of the paragraph:

1) Changed (page 10): “Over time, excess sympathetic activation combined with reduced parasympathetic influence may result in left ventricular remodeling and atherosclerosis in patients with hypertension, increasing the risk of microvascular (e.g., vascular dementia) and macrovascular (e.g., myocardial infarction) injuries [12].”

to 

“Over time, excess sympathetic activation combined with reduced parasympathetic influence may result in left ventricular remodeling and atherosclerosis due to increased inflammation potentially as a consequence of excess sympathetic activation in patients with hypertension, increasing the risk of microvascular (e.g., vascular dementia) and macrovascular (e.g., myocardial infarction) injuries [12].”

Page 13: “breaths/min–1” is incorrect. Please replace with “breaths⋅min–1” or “breaths/min”.

As per the reviewer’s request, we have made the following change in the last sentence of procedures in page 13:

1) Changed: “breaths/min-1”

to

“breathsꞏmin-1”

Page 14: Is peak oxygen consumption really a potential covariate of sport tester validity? How was it tested in this manuscript? I have found very limited information in the result section.

We thank the reviewer for this important question regarding the potential role of VO2peak as a covariate in agreement of R-R interval detection between Polar V800TM and 12-lead ECG. Our reason to include VO2peak as a covariate derives from the findings of CARDIO-FIT study by Pathak et al. (2015) Impact of CARDIOrespiratory FITness on Arrhythmia Recurrence in Obese Individuals with Atrial Fibrillation: The CARDIO-FIT Study. The authors argue that overweightness and obesity beget arrythmias, which are considered artifacts in the context of HRV pre-processing. Pathak et al. (2015) hypothesized a potential connection between cardiorespiratory fitness and arrythmia occurrence as the positive correlation between cardiorespiratory fitness and cardiac health is well-established in the literature. To test their hypothesis, they divided 308 overweight or obese adults into low, adequate, and high cardiorespiratory fitness groups based on baseline VO2peak and followed them for four years to measure their fitness levels' impact on arrhythmia recurrence. After 4-years of follow up, 84% in high aerobic capacity group no longer had arrhythmias, compared with 76% in the average and 17% in the low aerobic capacity groups. Our reason to include VO2peak as a potential covariate was merely to explore whether cardiorespiratory fitness status can impact the occurrence of artifacts among individuals with hypertension and thus ensure the agreement between Polar V800TM and ECG was not confounded by any external variable.

As far as the statistical testing of this variable as a potential covariate, we formed aerobic capacity subgroups based on fitness quartiles, separating the aerobic capacity group into 1) excellent aerobic fitness, 2) fair to good aerobic fitness, and 3) very poor to poor aerobic fitness according to the 10th edition of the American College of Sports Medicine’s Guidelines for Exercise Testing and Prescription (2018). All statistical analyses that were performed in all subjects (n=25) were also carried out for different subgroups of aerobic capacity. The results of these analyses can be seen as supporting information.

Page 14: The subtitle “R-R Interval Recording” is rather “R-R Interval Processing”. Please consider rephrasing.

As per the reviewer’s request, we have made the following changes in the subtitle in page 14:

1) Deleted: “R-R Interval Recording”

2) Added: “R-R Interval Processing”

Page 14: “BTC” should be replaced by “one of the authors (BTC)”. 

As per the reviewer’s request, we have made the following change in under R-R interval Processing subtitle of the methods in page 14: 

1) Changed: “BTC”

to 

“one of the authors (BTC)”

2) Added: “…checked by another author (DAG) to ensure data accuracy and…..”

Page 14: So, the correct position of the R peak was performed by only one author? Given that R-R intervals obtained from a medical ECG device were used as a gold standard in this manuscript, I consider that a single author checking is not sufficient.

We thank the reviewer for pointing out the missing information concerning the identification of correct R peak position. The first author of our paper (BTC) performed artifact identification as well as corrections first and then his findings were checked and confirmed by a co-author (DAG) who has published two validation papers at rest and during exercise using the Polar V800TM among healthy individuals. Giles D, Draper N, Neil W; Validity of the Polar V800 heart rate monitor to measure RR intervals at rest. Eur J Appl Physiol 2016;116(3):563-571; Giles DA, Draper N. Heart Rate Variability During Exercise: A Comparison of Artefact Correction Methods. J Strength Cond Res 2018 March 01;32(3):726-735. We, therefore, made the following changes in “Artifact Identification (pg 14)” and “Artifact Correction (pg 15)” sections to show that two authors were involved in the processing of HRV data:

1) Added (pg 14): “Artifact identification was performed by the first author (BTC) and each step of this process was checked by a co-author (DAG).”

Page 14: Please describe the MC method in more detail. Who performed it? In which kind of software? Did the person have RR intervals available from the medical ECG device or not?

We thank the reviewer for raising this concern regarding the details of the MC method. As stated previously, all artifact identification and correction processes were performed by the first author (BTC) and then confirmed by a co-author (DAG). Following the identification of artifacts as described under the “Artifact Identification” section of the methods, the artifacts were manually corrected following the guideline provided in Table 1 (Types, description, and correction methods of artifacts). The corrected R-R intervals were then imported into Kubios HRV Premium (ver.3.2) to obtain the HRV measures. All correction methods, including MC were used to correct artifacts present in the R-R intervals derived from the Polar V800TM and not from the ECG. The R-R intervals obtained from ECG were only corrected in the case of physiological artifacts (e.g., ectopic beats), which were detected on ECG strips. All physiological artifacts that were present in three subjects (N=7) were replaced by interpolated R-R intervals from adjacent R-R intervals. We, therefore, made the following changes in page 14 to describe the MC method in more details:

1) Added: “All artifact corrections were performed by the first author (BTC) and each step of this process was checked by a co-author (DAG).”

2) Deleted: “For the MC, non-sinus beats in both signals (N=3) were replaced by interpolated R-R intervals from adjacent R-R intervals.”

3) Changed: “Corrections were made only for T2, T3, T4, T5, and T6b since it is not possible to identify T1 and T6a artifacts without simultaneous ECG recording.”

to

“The MC was used only for T2, T3, T4, T5, and T6b artifacts present in R-R intervals derived from the Polar V800TM following the guidelines provided in Table 1. The T1 and T6a artifacts in the Polar V800TM R-R intervals, however, were not corrected since it is not possible to identify both artifacts without simultaneous ECG recordings.”

4) Added: “The corrected Polar V800TM R-R intervals by the MC were then imported into Kubios HRV Premium (ver.3.2) to calculate HRV measures. All non-sinus beats present in the ECG signal (N=7) were replaced by interpolated R-R intervals from adjacent R-R intervals.”

Page 16: “… in both signals (N = 3)” does it mean three occurrences of artifacts in all group of RR recordings? Or does it mean that recordings from three subjects contains artifacts?

We thank the reviewer for his comment and acknowledge the confusing statement about the number of non-sinus beats. We attempted to report the number of subjects with physiological artifacts, which were detected in the ECG strips of three subjects (thus N=3). However, we decided that reporting the total number of non-sinus beats (N=7) can be more revealing for the readers. Thus, we have made the following changes:

1) Deleted: ““For the MC, non-sinus beats in both signals (N=3) were replaced by interpolated R-R intervals from adjacent R-R intervals.”

2) Added: “The corrected Polar V800TM R-R intervals by the MC were then imported into Kubios HRV Premium (ver.3.2) to calculate HRV measures. All non-sinus beats present in the ECG signal (N=7) were replaced by interpolated R-R intervals from adjacent R-R intervals.”

Page 17: “unpredictable heartbeat dynamics” should be replaced by ”chaotic heartbeat dynamics”. A chaotic system is theorethically predictable, but it is a complex nonlinear system, so in practise the chaotic system looks stochastic (random). Mathematical theory dealing with nonlinear HRV indexes considers ANS to be chaotic, not stochastic. See, for example, Ernst G (2014). Heart rate variability. London: Springer.

We thank the reviewer for this insightful comment about the nature of the heartbeat dynamics. As per the reviewer’s suggestion, we have made the following change:

1) Changed: “Non-linear domain measures represent the unpredictable heartbeat dynamics caused by the complex interactions between several regulatory systems and they included: standard deviation 1 (SD1), SD2, and SD2/SD1 ratio. The non-linear measures were analyzed as a Poincaré plot, a graph in which each N-N interval is plotted against the next N-N interval, making a scatter plot [28]”

to

“Non-linear domain measures represent the chaotic heartbeat dynamics caused by the complex interactions between several regulatory systems. 

Page 17: I agree that Poincaré plot is one of the graphical methods used to study chaotic systems. Poincaré plot can therefore be considered as a nonlinear method. However, I cannot agree that SD1, SD2, and SD2/SD1 can be considered as nonlinear indexes of HRV. It was shown that these indexes can be calculated from SDNN and SDSD (Sassi R, Cerutti S, Lombardi F, Malik M, Huikuri H V, Peng C-K, … Yamamoto Y (2015). Advances in heart rate variability signal analysis: Joint position statement by the e-cardiology ESC working group and the european heart rhythm association co-endorsed by the asia pacific heart rhythm society. Europace, 17, 1341–1353). Therefore these indexes should be considered as time domain indexes. Sample entropy is nonlinear HRV index suitable for short-term (≈5 min) RR recordings. Therefore, I suggest removing the indexes SD1, SD2, and SD1/SD2 and adding sample entropy.

We thank the reviewer for providing evidence to show the connection between time domain measures of SDNN and SDSD and SD1, SD2, and SD/SD1. We agree with the potential rationale to remove SD1, SD2, and SD/SD1. Therefore, we have made the following changes: 

1) Changed (page 17): “Non-linear domain measures represent the chaotic heartbeat dynamics caused by the complex interactions between several regulatory systems, and they included: standard deviation 1 (SD1), SD2, and SD2/SD1 ratio.”

to

“Non-linear domain measures represent the chaotic heartbeat dynamics caused by the complex interactions between several regulatory systems.”

2) Added (page 17): “We included sample entropy (SampEn) as a measure of non-linear methods. SampEn measures the signal complexity or irregularity where large SampleEn suggests high irregularity and smaller SampleEn a more regular signal [11].”

3) Deleted (page 17): “The non-linear measures were analyzed as a Poincaré plot, a graph in which each N-N interval is plotted against the next N-N interval, making a scatter plot [28]. The analysis consisted of placing an ellipse to the plotted points [28]. The SD of the distance of each plotted N-N interval from the y= x axis (SD1 or ellipse’s width), the SD of each plotted N-N interval from the y = x + average N-N interval (SD2 or ellipse’s length), and SD2/SD1 ratio were then calculated [28].”

4) Changed (page 21): “The UN ES ranged from 0.091 to 0.835, indicating the magnitude of difference between the HRV measures calculated from UN Polar V800TM HR monitor and ECG R-R intervals was trivial to large, and had five out of 11 HRV domain measures between 0.2≤ and <0.5, indicating a moderate difference. Additionally, the UN ICCs ranged from 0.070 to 0.98 suggesting poor to excellent agreement, but had seven out of 11 HRV measures below <0.5, indicating poor agreement. Between the two correction methods of Kubios HRV Premium (ver. 3.2), the TBC resulted in smaller biases than the AC for SDNN, RMSSD, pNN50%, LFnu, HFnu, LF/HF ratio, SD1, SD2, and SD2/SD1 but the AC produced smaller biases than TBC for LFms2,HFms2 and SampEn (Tables 4 and 5). Additionally, when the TBC was compared to the AC method, the TBC produced tighter LoA ranges and smaller ES (<0.031 versus <0.085) for all HRV.”

to

“The UN ES ranged from 0.091 to 0.835, indicating the magnitude of difference between the HRV measures calculated from UN Polar V800TM HR monitor and ECG R-R intervals was trivial to large, and had five out of nine HRV domain measures between 0.2≤ and <0.5, indicating a moderate difference. Additionally, the UN ICCs ranged from 0.070 to 0.98 suggesting poor to excellent agreement, but had six out of nine HRV measures below <0.5, indicating poor agreement. Between the two correction methods of Kubios HRV Premium (ver. 3.2), the TBC resulted in smaller biases than the AC for SDNN, RMSSD, pNN50%, LFnu, HFnu, LF/HF ratio, but the AC produced smaller biases than TBC for LFms2,HFms2 and SampEn (Tables 4 and 5). Additionally, when the TBC was compared to the AC method, the TBC produced tighter LoA ranges and smaller ES (<0.031 versus <0.085) for all HRV domain measures except for SampEn.”

5) Deleted (page 22): 

SD1 (ms) 27.1±20.8 65.7±68.2 -38.62 (-158.55 to 81.31) 0.21 (-0.11-0.52) 0.765

SD2 (ms) 72.7±33.6 106.5±62.1 -33.80 (-137.14 to 69.53) 0.37 (-0.01-0.66) 0.677

SD2/SD1 Ratio 3.4±1.4 2.7±1.6 0.66 (-1.45 to 2.76) 0.68 (0.32-0.85) 0.442

UN Table 3

SD1 (ms) 27.1±20.8 25.6±18 1.51 (-18.12 to 21.14) 0.87 (0.73-0.94) 0.078

SD2 (ms) 72.7±33.6 75.6±36.4 -2.97 (-26.11 to 20.17) 0.94 (0.87-0.97) 0.085

SD2/SD1 Ratio 3.4±1.4 3.4±1.3 -0.07 (-1.01 to 0.87) 0.94 (0.86-0.97) 0.055

AC Table 4

SD1 (ms) 27.1±20.8 26.7±19.5 0.38 (-6.78 to 7.53) 0.98 (0.96-0.99) 0.019

SD2 (ms) 72.7±33.6 73.8±35.7 -1.09 (-14.39 to 12.21) 0.98 (0.96-0.99) 0.031

SD2/SD1 Ratio 3.4±1.4 3.3±1.4 0.03 (-0.74 to 0.80) 0.96 (0.91-0.98) 0.021

TBC Table 5

SD1 (ms) 27.1±20.8 27.0±20.6 0.13 (-1.29 to 1.54) 1.00 (1.00-1.00) 0.006 

SD2 (ms) 72.7±33.6 72.5±33.5 0.21 (-0.55 to 0.97) 1.00 (1.00-1.00) 0.006 

SD2/SD1 Ratio 3.4±1.4 3.4±1.4 0.01 (-0.14 to 0.15) 1.00 (1.00-1.00)0.005 

MC Table 6

6) Added: 

SampEn 1.5±0.2 1.2±0.5 0.33± (-0.58 to 1.24) 0.39 (-0.18 to 0.71) 0.836

UN Table 3

SampleEn: Sample Entropy

SampEn 1.5±0.2 1.5±0.3 0.02 (-0.21 to 0.21) 0.81 (0.57-0.91) 0.039

AC Table 4

SampEn 1.5±0.2 1.5±0.3 0.02 (-0.35 to 0.40) 0.93 (0.85-0.97) 0.039

TBC Table 5

SampEn 1.5±0.2 1.5±0.2 0.01 (-0.11 to 0.12) 0.97 (0.93-0.99) 0.001

MC Table 6

Page 18: ”inter correlation coeficient” should be replaced by “intraclass correlation coefficient”.

A per the reviews request, we have made the following change:

1) Changed: “The inter class correlation coefficient (ICC) with the 95% confidence interval (CI) assessed the concurrent validity (or interchangeable agreement) of the R-R intervals and HRV measures [33].”

to 

“The intraclass correlation coefficient (ICC) with the 95% confidence interval (CI) assessed the concurrent validity (or interchangeable agreement) of the R-R intervals and HRV measures [33].”

Page 18: What type of ICC was used? There are several types of ICC. Please specify the ICC you used according to Koo & Li (2016) or a similar source.

We thank the reviewer for this comment. We followed the ICC guideline published Koo & Li (2016) to choose appropriate type for our study. We specify the type as well as model and definition of ICCs reported in our study as recommended by the authors. Type: average measures (averaged measures of k raters for each subject) that shows agreement across the raters; model: two way-mixed that assumes a random effect of subjects but a fixed effect of rater; and definition: absolute agreement that shows whether two HRV scores from the Polar V800TM and ECG exactly match. Therefore, we have made the following change in page 18 regarding the type, model, and definition of the ICC analysis in our study: 

1) Added: “We selected a two-way mixed model with absolute agreement and reported average measures (mean of two measurements).”

Page 18: You did not specify which variables were treated by the logarithmic transformation. Please add this information.

We thank the reviewer for this comment. As per the reviewer’s comment, we have made the following change in page 18:

1) Changed: “In the case of heteroscedasticity in the HRV domain measures, the data were logarithmically transformed before the calculation of LoA ranges.”

to 

“The RMSSD, pNN50%, LFms2, and HFms2 were logarithmically transformed before the calculation of LoA ranges.”

Page 18/19: “The main findings of gender, BMI, medication use, and aerobic capacity sub-groups were similar to those in all subjects. Moreover, the removal of subjects using B-blockers from the analyses did not have an impact on the findings.” Please be more specific or remove these sentences.

We thank the reviewer for this comment. We aimed to study the potential impacts of gender, BMI, medication, and aerobic capacity by separating these variables into sub-groups of male and female (gender), overweight and obese (BMI), medication, non-medication (medication use) and very poor to poor, fair to good, and excellent (aerobic capacity). All statistical analyses that were performed in all subjects were also performed for different subgroups of gender, medication use, BMI, and aerobic capacity. The results of our sub-group analyses as well as the individual data, which the results were based on can be seen under “Supporting Information”. Moreover, we have made the following changes in “Statistical Analyses” and “Results” sections to clarify the role of gender, BMI, medication use, and aerobic capacity on the agreement of the Polar V800TM and ECG in producing R-R intervals:

1) Changed (statistical analyses section): “Finally, different sub-groups of gender, BMI, medication use (particularly β-Blockers), and aerobic capacity were also analyzed using the same statistical analyses in the main findings to rule out their potential impacts on the main findings.”

to

“Finally, the sub-group analyses of gender, BMI, medication use, and aerobic capacity were performed using the same statistical methods as in all subjects to study their potential impacts on the agreement between Polar V800TM and ECG. Gender and medication use were grouped into “male and female” and “medication and non-medication”. Further, BMI and aerobic capacity were divided into “overweight (25.0-29.9 kg/m2)” and “obesity (>30 kg/m2)” and “very poor to poor (15.6-37.5 ml/kg/min)”, “fair to good (32.0-41.5 ml/kg/min)”, and “excellent (48.6-51.7 ml/kg/min)”, respectively [25].”

2) Deleted (results): “The main findings of gender, BMI, medication use, and aerobic capacity sub-groups were similar to those in all subjects. Moreover, the removal of subjects using B-blockers from the analyses did not have an impact on the findings.”

3) Added (results):“Agreement of HRV Measures Separated by Medication Use, Gender, BMI, and Aerobic Capacity (title)”

4) Added (results): “The magnitude of biases, LoA ranges, ICCs, and effect sizes of HRV measures of the subjects with medication use S1A-S1D Tables, males S2A-S2D Tables, those with overweight BMI S3A-S3D Tables, and those with very poor-to-poor aerobic capacity S4A-S4D Tables were similar to those of non-medication, female, obesity, and excellent aerobic capacity between the Polar V800TM and ECG. Also, as was in all subjects, the correction of artifacts substantially improved the disagreement seen between UN HRV measures of the Polar V800TM and ECG. Moreover, the accuracy of artifact correction methods in correcting R-R intervals were similar to those reported in all subjects (AC<TBC≤AC) S4-S7 Datasets.”

Page 19, Table 2: There is a significant difference in systolic blood pressure. However, no statistical procedure is described in the statistical analysis subsection. Is it a two-sample t-test?

We thank the reviewer for pointing out the missing statistical information. As per the reviewer’s suggestion we have made the following changes to clarify the statistical method used to investigate the group differences between males and females:

1) Added: “The independent sample t-test was used to report the mean differences between the subject characteristics of men and women.”

Page 19: What does “Agreeability and interchangeability” mean? I suggest to use the term agreement. 

As per the request of the reviewer we have made the following change throughout our study:

1) Changed “Agreeability and interchangeability” 

to 

“Agreement” 

Page 19: Why is the average length of the HRV measurement 4.6±0.9 min? I expect a value of 5 min.

We thank the reviewer for his comment. We aimed to obtain 5 min of HRV measurement but unfortunately, the signal from the chest strap to heart rate monitor was interrupted in eight subjects for unknown reasons, which made the total measurement duration less than 5 min. We have made the following changes to clarify the reason for 4.6 min HRV measurement:

1) Added (results): “The length of HRV measurement for eight subjects was less than 5 min due to the loss of connection between the HR monitor and the strap for unknown reasons.”

2) Changed (results): “The average length of the HRV measurement was 4.6±0.9 min and number of R-R intervals was 333.0±70.5.”

to

“Therefore, the average length of the HRV measurement was 4.6±0.9 min and number of R-R intervals was 333.0±70.5.”

Page 19, Table 6: The results of 1.00 (1.00 – 1.00) for ICC (95% CI) look strange because ICC = 1 means a perfect agreement that cannot be reached in practise. In your manuscript, the ICC could not be equal to 1 because Figure 1D shows several discrepancies and the associated LoAs are not zero. Maybe it is due to rounding. Please try to write the ICC value for more decimal digits if the ICC values I close to 1.

We thank the reviewer for his comment. The reason for the discrepancies seen on Figure 1D was because we did not correct T1 and T6a errors as neither error is visible without simultaneous ECG recording. We have made the following change to explain the discrepancies seen on Figure1D:

1) Added (discussion, paragraph 3): “Of note, the Polar V800TM uncorrected T1 and T6a errors can be seen on the corrected Bland–Altman plots (Figure 1D) as outliers, a similar finding reported by Giles et al. [22]. T1 and T6 errors are not visible without simultaneous ECG recording.”

We obtained an exact ICC score of 1.00 (1.00-1.00) for many HRV variables produced especially by MC R-R intervals. We repeated the ICC analyses but still received the same results of 1.00 (1.00-1.00), particularly for MC HRV results. We can explain the reason for the ICCs of 1.00 (1.00-1.00) based on the McGraw et al. (1996) paper. These authors define the intra-class correlation as the ratio of the variance attributable to subjects divided by the total of subject and error variances. They argue that if the ratings by the raters are in perfect agreement, which appears to be the case especially when the artifacts were corrected by MC in our study then there will not be any within-subject variation and thus no error variance. Consequently, that ratio will involve a variance term for subjects divided by a variance term for subjects plus a variance term for error, making the latter exactly 0. Therefore, the ratio will turn out 1.00 (1.00-1.00) based on the aforementioned definition of ICC. McGraw, K. O., & Wong, S. P. (1996). Forming inferences about some intraclass correlation coefficients. Psychological Methods, 1(1), 30–46.

---

## [Decision Letter · Decision Letter 1]

23 Sep 2020

Validation of the Polar V800TM Heart Rate Monitor and Comparison of Artifact Correction Methods among Adults with Hypertension

PONE-D-20-14879R1

Dear Dr. Cilhoroz,

We’re pleased to inform you that your manuscript has been judged scientifically suitable for publication and will be formally accepted for publication once it meets all outstanding technical requirements.

Please, address the minor issues raised by the reviewer during the proofreading.

Kind regards,

Daniel Boullosa

Academic Editor

PLOS ONE

Additional Editor Comments (optional):

Reviewers' comments:

Reviewer's Responses to Questions

**Comments to the Author**

1. If the authors have adequately addressed your comments raised in a previous round of review and you feel that this manuscript is now acceptable for publication, you may indicate that here to bypass the “Comments to the Author” section, enter your conflict of interest statement in the “Confidential to Editor” section, and submit your "Accept" recommendation.

Reviewer #1: All comments have been addressed

2. Is the manuscript technically sound, and do the data support the conclusions?

Reviewer #1: Yes

3. Has the statistical analysis been performed appropriately and rigorously? 

Reviewer #1: Yes

4. Have the authors made all data underlying the findings in their manuscript fully available?

Reviewer #1: Yes

5. Is the manuscript presented in an intelligible fashion and written in standard English?

Reviewer #1: Yes

6. Review Comments to the Author

Reviewer #1: I have read the revised manuscript and the response to reviewers. I appreciate that the authors provided raw data as supplementary material and added sample entropy to the results. I performed my own ICC calculations and got the same results as the authors. It is still debatable that 1.00 (1.00–1.00) is the correct notation. Now I consider this acceptable, because it is, for example, 0.999396 (0.998646 ± 0.999733) rounded to two decimal places.

I am satisfied with the manuscript and authors’ responses. I have no further request.

I have two typographic notes.

1) On page 24, there is “breaths min-1” where there is no dot between ”breaths” and “min”. The dot is probably visible in Word processor but lost in the pdf file.

2) Please unify the style of the units in the whole manuscript. As an example, please compare kg.m-2 in Table 2 and kg/m2 in the text above Table 2.

7. PLOS authors have the option to publish the peer review history of their article (what does this mean?). If published, this will include your full peer review and any attached files.

Reviewer #1: **Yes: **Jakub Krejčí

---

## [Editor Report · Acceptance letter]

29 Sep 2020

PONE-D-20-14879R1 

Validation of the Polar V800 Heart Rate Monitor and Comparison of Artifact Correction Methods among Adults with Hypertension 

Dear Dr. Cilhoroz:

I'm pleased to inform you that your manuscript has been deemed suitable for publication in PLOS ONE. Congratulations! Your manuscript is now with our production department. 

Kind regards, 

on behalf of

Dr. Daniel Boullosa 

Academic Editor

PLOS ONE